# In Pursuit of Optimal Quality: Cultivar-Specific Drying Approaches for Medicinal Cannabis

**DOI:** 10.3390/plants13071049

**Published:** 2024-04-08

**Authors:** Matan Birenboim, Nimrod Brikenstein, Danielle Duanis-Assaf, Dalia Maurer, Daniel Chalupowicz, David Kenigsbuch, Jakob A. Shimshoni

**Affiliations:** 1Department of Food Science, Institute for Postharvest and Food Sciences, Agricultural Research Organization, Volcani Center, P.O. Box 15159, Rishon LeZion 7505101, Israel; 2Department of Plant Science, The Robert H Smith Faculty of Agriculture, Food and Environment, The Hebrew University, Rehovot 7610001, Israel; 3Department of Postharvest Science, Institute for Postharvest and Food Sciences, Agricultural Research Organization, Volcani Center, Rishon LeZion 7505101, Israel

**Keywords:** *Cannabis sativa* L., controlled drying, atmospheric drying, cannabinoids, terpenes

## Abstract

A limited number of studies have examined how drying conditions affect the cannabinoid and terpene content in cannabis inflorescences. In the present study, we evaluated the potential of controlled atmosphere drying chambers for drying medicinal cannabis inflorescence. Controlled atmosphere drying chambers were found to reduce the drying and curing time by at least 60% compared to traditional drying methods, while preserving the volatile terpene content. On the other hand, inflorescences subjected to traditional drying were highly infested by *Alternaria alternata* and also revealed low infestation of *Botrytis cinerea*. In the high-THC chemovar (“240”), controlled N_2_ and atm drying conditions preserved THCA concentration as compared to the initial time point (t_0_). On the other hand, in the hybrid chemovar (“Gen12”) all of the employed drying conditions preserved THCA and CBDA content. The optimal drying conditions for preserving monoterpenes and sesquiterpenes in both chemovars were C5O5 (5% CO_2_, 5% O_2_, and 90% N_2_) and pure N_2_, respectively. The results of this study suggest that each chemovar may require tailored drying conditions in order to preserve specific terpenes and cannabinoids. Controlled atmosphere drying chambers could offer a cost-effective, fast, and efficient drying method for preserving cannabinoids and terpenes during the drying process while reducing the risk of mold growth.

## 1. Introduction

*Cannabis sativa* L., the sole species in the *Cannabaceae* family, is an annual herb with proven therapeutic benefits for conditions like pain, epilepsy, and cancer, among others [1,2,3,4]. Despite various available cannabis products, dried inflorescences are among the prevalent medicinal cannabis products accessible to patients. The plant’s therapeutic effects are attributed mainly to two major secondary metabolite classes, the cannabinoids and terpenes [5,6]. The cannabinoids interact with the body’s endocannabinoid system, influencing various physiological processes [6,7]. The most studied cannabinoids are (−)-Δ9-trans-tetrahydrocannabinol (THC) and cannabidiol (CBD) [8]. THC is known for its psychoactive effects but also provides medicinal benefits such as reducing chronic pain, stimulating appetite, and proving beneficial in conditions like Alzheimer’s disease and cancer [2,9]. CBD, a non-psychoactive cannabinoid, is recognized for its anti-inflammatory, anxiolytic, and antiepileptic properties [8,9,10]. Terpenes, the volatile aromatic compounds found in the inflorescence of medicinal cannabis, play a pivotal role in enhancing cannabis therapeutic efficacy and consumer experience [9,11]. This aspect of consumer experience is crucial in medicinal cannabis formulations, as it aids in consistent patient adherence to treatment regimens, ultimately impacting the effectiveness of the therapy and patient compliance [11,12]. Scientific research has highlighted the importance of terpenes in the “entourage effect”, a synergistic interaction where terpenes, in conjunction with cannabinoids like THC and CBD, enhance the overall therapeutic potential of cannabis [9,13]. This interaction potentially amplifies the analgesic, anti-inflammatory, and anxiolytic properties of medicinal cannabis, thus contributing to its clinical effectiveness [9]. Furthermore, terpenes have been individually noted for their medicinal properties. For example, β-myrcene possesses anti-inflammatory and analgesic qualities, while d-limonene and linalool are known for their anxiety-reducing and antidepressant effects [9]. The complexity and variability of terpene profiles in different cannabis cultivars underscore the importance of employing optimal drying, curing, and storage techniques to preserve consistent terpene composition during post-harvest processes, which is paramount for ensuring patient satisfaction and therapeutic consistency.

Cannabis cultivars are categorized into three main classes based on the ratio of the major cannabinoids (−)-Δ9-trans-tetrahydrocannabinolic acid (THCA) and cannabidiolic acid (CBDA) and their neutral homologs THC and CBD: high-THCA (total THC/total CBD ratio ≥ 10), high-CBDA (total CBD/total THC ratio ≥ 10), and hybrid (10 > total THC/total CBD ratio > 0.1) [14,15,16,17,18]. These acidic cannabinoids convert to their neutral forms by a decarboxylation process under specific post-harvest conditions such as light intensity and temperature [19,20,21]. Given the chemical variability within and between cultivars, the term “chemovar”, encompassing the full cannabinoid and terpene profile, is preferred for classification [7,15,19].

Medicinal cannabis inflorescences undergo rigorous post-harvest processes to ensure optimal quality [1,22]. Initial stages include trimming and a 2–3 week drying period, influenced by factors like temperature and humidity [1,19,21,23]. This processing stage results in up to 75% weight loss, with slight variations due to environmental conditions and different chemovars [19,24]. Drying can be conducted by hanging buds or using trays, though the latter has drawbacks like increased mold susceptibility [19,23,25]. Final drying, or curing, often involves sealing the inflorescences in containers and periodically airing them to remove residual moisture before final packaging [1,25].

A limited number of studies have examined how drying conditions affect the cannabinoid and terpene content in cannabis inflorescences [19,21,23,24,25,26,27,28,29]. Recent literature suggests that industrial drying techniques have seen little innovation and are often more art than science, emphasizing the need for empirically grounded, efficient methods [21,23,25]. Recent reports have explored various alternative drying methods, including microwave, convection, freeze-drying, infrared, non-isothermal, and vacuum drying techniques [19,21,23,24,25,26,27,28,29]. However, these methods have shown limitations in effectively preserving the volatile terpenes, necessitating additional research. Future investigations are also required to establish consistent results across diverse cannabis chemovars, and to ensure the retention of a broader range of cannabinoids and terpenes.

Controlled atmosphere drying uses controlled atmosphere chambers which allow for controlling the temperature, humidity, and gas composition during the drying process [30,31,32]. The drying or storing of fruits and vegetables in controlled atmosphere chambers under a controlled gas environment has been shown to increase the produce’s shelf-life, increase produce quality over time, and preserve the volatile composition [30,31,32,33]. Hence, controlled atmosphere drying could potentially offer a superior solution for cannabis inflorescence drying in terms of both speed and the preservation of terpene and cannabinoid content [30,31].

This research aimed to determine the optimal atmospheric composition within controlled atmosphere chambers to facilitate the swift drying of two specific commercially available cannabis inflorescence chemovars, with a focus on preserving their distinct terpene and cannabinoid content.

## 2. Results and Discussion

### 2.1. 240 and Gen12 Initial Chemical Composition Comparison

The chemovar with high THCA content, 240, exhibited initial THCA levels approximately two times greater than those found in the Gen12 hybrid chemovar (Table 1). In contrast, the Gen12 chemovar demonstrated initial CBDA levels that were two orders of magnitude higher than in the 240 chemovar (Table 1). For minor cannabinoids (below 1 DW%), the Gen12 chemovar revealed significantly higher levels of CBGA, CBG, and CBCA than the 240 chemovar, yet both chemovars revealed similar levels of THC (Table 1). Furthermore, only the 240 chemovar had detectable levels of THCVA and (−)-Δ9-trans-tetrahydrocannabiorcolic-C4 acid (THCA-C4), while CBD and CBDVA were unique to the Gen12 chemovar (Table 1).

β-myrcene and d-limonene were the primary monoterpenes in the Gen12 and 240 chemovars, respectively (Table 2). However, the monoterpene content in the 240 chemovar was significantly lower compared to the Gen12 chemovar (Table 2). The Gen12 chemovar had one order of magnitude higher β-myrcene and (−)-β-pinene levels and two orders of magnitude higher α-pinene levels as compared to the 240 chemovar at t_0_ (Table 2). Furthermore, the Gen12 chemovar had 2.5-fold higher initial sesquiterpene content compared to the 240 chemovar (Table 2). Though both chemovars had β-caryophyllene as the dominant sesquiterpene, Gen12’s concentration was two times higher than that of the 240 chemovar at t_0_ (Table 2). Nerolidol, α-guaiene, α-bulnesene, α-gurjunene, and elemol were the only sesquiterpenes found in higher concentrations in the 240 chemovar (<0.1 DW%, Table 2).

### 2.2. Drying Process Efficiency

After six days, all the cannabis inflorescences subjected to the three distinct controlled atmosphere drying conditions were found to be completely dried. This conclusion was drawn from the absence of weight change on the sixth day, indicating that no further curing was necessary. On the other hand, the cannabis inflorescences in the reference group from both chemovars failed to reach a final stable dry weight value after 14 days of open-air drying, indicating that excessive water content was still present (Table 3). Consequently, an additional day of curing was necessary, employing silica gel pearls humidity absorbers, to achieve completely dried cannabis inflorescences (Table 3). After 14 days of drying in the open air, the 240 and Gen12 inflorescences contained 45% and 16% of water, respectively, as compared to the 240 and Gen12 inflorescences dried under CA drying conditions. This study’s findings reveal that drying cannabis inflorescences under controlled atmosphere drying conditions results in a process that is 2.5 times faster than open-air traditional drying. It is important to highlight that the traditional method of cannabis curing involves large, manually-ventilated containers where the complete removal of the remaining water can span two to three weeks [1]. Hence, controlled atmosphere drying chambers substantially speed up the drying process compared to the methods typically used today.

Prolonged drying can elevate mold risks due to slow moisture removal which allows mold to promote growth, crucial for medicinal cannabis [19]. In this study, no visible molds were detected after drying in all inflorescences from both chemovars subjected to CA drying chambers. On the other hand, approximately 80% of the 240 chemovar inflorescences subjected to traditional drying were highly infested by *Alternaria alternata* and also revealed low infestation of *Botrytis cinerea* on day 12 of the drying process (Figure 1 and Table 4). In order to assess the infestation level in all samples, both CFU and real-time PCR assays were conducted (Table 4). The extent of infestation with *Alternaria alternata* after CA drying was higher but not statistically significant as compared to the infestation level of fresh inflorescence (Table 4). Moreover, the infestation with *Botrytis cinerea* at t_0_ and after CA drying was below the limit of detection and therefore reported as undetermined (Table 4). On the other hand, *Alternaria alternata* infestation levels and CFU levels of inflorescences subjected to traditional drying were higher by one to two orders of magnitude as compared to samples subjected to CA drying conditions or at t_0_ (Table 4). Therefore, CA drying hastened the drying process and delayed mold growth in cannabis inflorescences, due to the lower duration of available water content conducive to mold. Regarding Gen12, the presence of *Botrytis cinerea* and *Alternaria alternata*, both at the initial time point (t_0_) and post-drying, was found to be below the limit of detection.

### 2.3. Impact of the Drying Process on the Cannabinoid Content Preservation

#### 2.3.1. High-THCA Chemovar—240

The cannabinoid content within the 240 chemovar was notably influenced by the different drying conditions (Figure 2 and Table 1). Specifically, the predominant cannabinoid—THCA—exhibited a concentration reduction (by 2–20%, see detailed explanation for relative concentration calculation in Section 3.7) throughout the different drying procedures implemented (Figure 2a and Table 1). The smallest reduction in THCA levels was observed under N_2_ drying conditions followed by atm (both are not statistically significant from t_0_, Figure 2a and Table 1). On the other hand, open-air drying conditions led to the lowest THCA concentration (Figure 2a and Table 1). Although the atmospheric composition of the controlled atmosphere chamber and open-air conditions were identical, they produced significantly different cannabinoid concentrations (Table 1). This discrepancy may be attributed to factors such as the duration of the drying process, the inclusion of silica gel pearls humidity absorbers, the use of dry gases within the controlled atmosphere chamber, and/or to the high mold infestation observed in open-air samples (Table 4). CBGA demonstrated a similar degradation pattern to THCA (Figure 2b). CBGA concentration remained stable during drying under N_2_ and atm drying conditions (not statistically significant from t_0_), while the most pronounced decrease was observed under open-air drying conditions (by 42%, Figure 2b and Table 1). CBGA, being the first acidic cannabinoid produced by the cannabis plant, serves as the primary cannabinoid transformed by three individual FAD-dependent dehydrogenases into THCA, CBDA, and CBCA [6,9]. This enzymatic activity may persist during the initial drying process at stages where the inflorescences remain sufficiently moist [34]. The decline in CBGA concentration did not yield increased levels of these three downstream acidic cannabinoids, nor an increase in CBG concentration due to ongoing CBGA decarboxylation (Figure 2a,c,d,h).

Under open-air drying conditions, CBDA concentration dropped by 24%, whereas under all other drying conditions, CBDA concentration remained stable (Figure 2c and Table 1). Varying decarboxylation patterns of THCA to THC were observed between the different drying conditions (Figure 2a,g). THC concentration under the controlled atmosphere drying conditions increased by 40–70% (Figure 2g and Table 1). In contrast, open-air drying conditions, which yielded the lowest post-drying THCA concentration, resulted in approximately 3.5-fold higher THC levels as compared to t_0_ (Figure 2a,g and Table 1). Notwithstanding, this increase in THC concentration during drying is still one order of magnitude lower as compared to the amount of THCA degraded during drying under open-air drying conditions (Figure 2a,g and Table 1). This observation may imply that controlled atmosphere drying conditions could either slow down the decarboxylation of THCA to THC or that the extended drying duration under open-air drying conditions allowed THCA decarboxylation to occur over a longer period, resulting in elevated THC levels. Total minor cannabinoid content after drying remained similar to t_0_, with the highest content observed for N_2_ drying conditions, especially in cannabinoids like CBGA, CBDA, THCVA, and CBG (Figure 2b,c,e,h,i). Consequently, total cannabinoid content was highest under N_2_ drying conditions, and lowest under open-air drying conditions (Figure 2i).

#### 2.3.2. Hybrid Chemovar—Gen12

The cannabinoids in the Gen12 chemovar displayed different drying patterns as compared to the 240 chemovar (Figure 3). Gen12 is a hybrid chemovar containing relatively high concentrations of both THCA and CBDA (Figure 3a,c). Unlike the degradation pattern of the most predominant cannabinoids observed in the 240 chemovar, in Gen12, stable levels were observed for the most predominant cannabinoids—THCA and CBDA under all drying conditions (Figure 3a,c). The relative changes compared to t_0_ for both THCA and CBDA under all drying conditions were smaller than 10% and not statistically significant from t_0_ (Figure 3a,c and Table 1). The stability of primary cannabinoids can differ notably between chemovars under identical drying conditions. This variation might stem from distinct genetic compositions, leading to significant disparities in enzyme activities, especially those associated with cannabinoid metabolism during drying [9].

CBGA concentration declined significantly under the different drying conditions, with the largest decrease observed under open-air drying conditions (by 37%, Figure 3b and Table 1) and the lowest concentration reduction observed under atm drying conditions (by 11%, not statistically significant from t_0_, Figure 3b and Table 1). Despite significant CBGA degradation occurring under all drying conditions, stable THCA, CBDA, CBCA, and CBG concentrations were observed (Figure 3a,c,d,f). Based on the known CBGA metabolic pathway, we anticipated seeing a corresponding increase in at least one of the CBGA metabolites following CBGA degradation. However, such a mass balance was not observed. Therefore, we speculate that an additional, unidentified degradation product may have been formed. Furthermore, the three controlled atmosphere drying conditions maintained stable THC and CBD concentrations, while under open-air drying conditions, approximately eight- and three-fold higher THC and CBD levels, respectively, were observed as compared to t_0_ (Figure 3e,g and Table 1). These results strengthen the hypothesis that controlled atmosphere drying conditions slow down the decarboxylation of the major cannabinoids or that the observed increase in neutral cannabinoids (THC and CBD) became possible due to the longer duration of the drying process under open-air drying conditions. Interestingly, in the Gen12 chemovar, the concentrations of THC and CBD increased under open-air drying conditions without any corresponding decrease in THCA and CBDA concentrations (Figure 3a,c,e,g). This suggests that the formation and degradation of cannabinoids continue to occur during the drying process.

The controlled atmosphere drying conditions maintained stable total minor cannabinoids concentration (not statistically significant from t_0_, Figure 3h). The highest total minor cannabinoids concentration was observed under open-air drying conditions (25% relative increase compared to t_0_), due to the increase in THC and CBD concentrations (Figure 3h). Moreover, none of the drying conditions yielded statistically significant differences in total cannabinoid content as compared to t_0_ (Figure 3i). In both chemovars, an increase in isomerization or oxidation degradation products such as Δ8-THC or CBN was not observed.

Uziel et al. recently found that both traditional and microwave oven drying of a high-THCA chemovar led to the decarboxylation of THCA to THC during the drying process (*p* < 0.001) [19]. In contrast, our study revealed that only traditional drying conditions resulted in significant decarboxylation (>3-fold and *p* < 0.0001) of major cannabinoids. Given that microwave drying is quicker than controlled atmosphere drying [19], this finding reinforces our conclusion that controlled atmosphere drying conditions are ideal for slowing the decarboxylation of THCA in high-THCA chemovars and the decarboxylation of THCA and CBDA in hybrid chemovars. Additionally, certain cannabinoids, such as THCA, CBGA, and THC, were found to have higher concentrations under various drying conditions [19], paralleling the varied cannabinoid concentration patterns observed in our study across different drying conditions used. Oduola et al. recently demonstrated that hot air, infrared, and microwave drying significantly reduce the drying time; however, they led to a substantial reduction of more than 45% in CBDA concentration, CBGA concentration, total CBD content, and total cannabinoid content compared to fresh inflorescence in all methods examined [26]. In a separate study, Chen et al. noted that hot air drying significantly shortened the drying time by one to two orders of magnitude in three high-CBDA chemovars [24]. However, it led to a reduction in total CBD content by 1.5–16%, depending on the drying temperature, compared to traditional drying and freeze-drying, both of which produced similar total CBD content [24]. The recent literature together with the results obtained from our study highlight the need for a drying procedure that will shorten the drying time without reducing the active compound content.

In summary, considering both the current findings and previous research, different chemovars, characterized by distinct secondary metabolite compositions and thus unique genotypic characteristics, can respond differently to various drying conditions, therefore necessitating chemovar-tailored drying conditions. These differences may be attributed to the varied kinetic behavior of enzymes involved in the cannabinoid biosynthetic pathways.

### 2.4. Impact of the Drying Process on the Terpene Content Preservation

#### 2.4.1. High-THCA Chemovar—240

Throughout the drying process, the formation or degradation patterns of monoterpenes varied among the different drying conditions. The highest concentrations of various monoterpenes in the 240 chemovar inflorescence were observed under C5O5 drying conditions (Figure 4). Specifically, (−)-β-pinene concentration remained stable under all drying conditions, except for C5O5, which led to a 18% relative concentration increase (Figure 4b and Table 2). Conversely, β-myrcene was highly sensitive to the drying process, losing approximately 3–20% of its initial concentration during drying, with the smallest reduction observed under C5O5 drying conditions (not statistically significant from t_0_) and the largest reduction under atm and open-air drying conditions (Figure 4c and Table 2). On the other hand, α-pinene and d-limonene concentrations remained stable during drying (Figure 4a,d and Table 2). 

During the drying process, the formation or degradation patterns of sesquiterpenes were similar among the different drying conditions. No particular drying conditions were found to be preferable for the various sesquiterpenes, as all conditions resulted in a 20–50% relative increase in their concentrations (Figure 5). β-caryophyllene and α-humulene revealed similar drying patterns under all drying conditions (Figure 5a,b and Table 2), a finding consistent with our prior study that demonstrated a high correlation in concentrations of these two sesquiterpenes across 14 different chemovars [18]. 

Taken altogether, the monoterpene content remained stable during drying with the highest monoterpene content observed under C5O5 drying conditions (not statistically significant from t_0_, Figure 6a and Table 2). Overall, sesquiterpene content significantly increased by 30–40%, with the highest increase observed under atm and N_2_ drying conditions, as compared to t_0_ (Figure 6b and Table 2). Although open-air samples were highly infested by *Alternaria alternata* and also revealed low infestation of *Botrytis cinerea*, their sesquiterpene composition was similar to controlled atmosphere samples. The total terpene content, encompassing both monoterpenes and sesquiterpenes, exhibited an increase of approximately 25–30% during the drying process (Figure 6c). Notably, atm and N_2_ drying conditions displayed a statistically significant enhancement in total terpene content compared to t_0_ (Figure 6c and Table 2).

Recent studies have revealed that certain terpenes, such as β-myrcene and d-limonene, are susceptible to degradation through freeze-drying in some chemovars [28,29]. These studies have also reported that the concentrations of certain terpenes, such as α-pinene, (−)-β-pinene, β-caryophyllene, and α-humulene, can increase under specific drying conditions [28,29]. Moreover, hot air drying was found to reduce various monoterpene concentrations by 38–95% while simultaneously increasing various sesquiterpene concentrations by 210–290% as compared to fresh inflorescence [26]. These results align with our observations indicating an increase in both terpene and cannabinoid concentrations during the drying process. This suggests that biological activities continue to occur within the cannabis inflorescences throughout the drying period.

#### 2.4.2. Hybrid Chemovar—Gen12

The highest concentrations of monoterpenes in the Gen12 inflorescence were observed under C5O5 drying conditions (Figure 7). Specifically, α-pinene concentration increased by 7–37%, while (−)-β-pinene concentration increased by 8–46% (Figure 7a,b and Table 2). Regarding these two specific terpenes, C5O5 drying conditions revealed statistically significant differences when compared to the other drying conditions and t_0_ (Figure 7a,b and Table 2). β-myrcene lost 16–30% of its initial concentration under atm, N_2_, and open-air drying conditions, whereas drying under C5O5 drying conditions yielded a 15% relative concentration increase (Figure 7c and Table 2). Consequently, C5O5 drying conditions were the best for β-myrcene and (−)-β-pinene preservation in both studied chemovars (Figure 4 and Figure 7 and Table 2). d-Limonene exhibited a relative concentration reduction ranging from 15–58%, with the smallest reduction observed under C5O5 drying conditions (not statistically significant from t_0_, Figure 7d and Table 2). In contrast, linalool concentration remained stable throughout the drying process, except under C5O5 drying conditions, which led to a notable 70% relative increase (Figure 7e and Table 2).

In the Gen12 chemovar, the concentration of seven of the fourteen sesquiterpenes, namely β-caryophyllene, α-humulene, (−)-guaiol, (−)-α-bisabolol, eudesma-3,7(11)-diene, γ-eudesmol, and β-eudesmol, increased during the drying process (Figure 8a–g). Notably, the open-air drying conditions produced a greater increase in their concentrations compared to all controlled atmosphere drying conditions (Figure 8a–g). Among these seven sesquiterpenes, β-caryophyllene and α-humulene demonstrated the smallest relative increase (21%) under open-air drying conditions (Table 2). These two sesquiterpenes also exhibited similar drying patterns under the other conditions examined, reinforcing the observed strong correlation between them (Figure 8a,b and Table 2). The other five sesquiterpenes previously mentioned exhibited a relative increase ranging from 42–59% under open-air drying conditions (Figure 8c–g and Table 2). In contrast, the highest relative increase observed for these sesquiterpenes under all controlled atmosphere drying conditions was 33% (Figure 8c–g and Table 2). Under open-air drying conditions, the concentration of bulnesol remained similar to t_0_, whereas all controlled atmosphere drying conditions resulted in a relative concentration increase of 30–39% (Figure 8h and Table 2). In the case of γ-gurjunene, β-eudesmene, α-selinene, β-bisabolene, and cis-α-bisabolene, the highest concentrations were achieved under N_2_ drying conditions (19–30% relative increase, Figure 8i,k–n). γ-elemene was the only sesquiterpene that displayed a concentration reduction, ranging between 8–18%, across all drying conditions (Figure 8j and Table 2). Notably, the γ-elemene concentration reduction observed under open-air drying conditions was the only one to show a statistically significant difference when compared to t_0_ (Figure 8j and Table 2).

An analysis of the overall monoterpene content in the Gen12 chemovar revealed that all drying conditions yielded a relatively stable monoterpene content, with a relative statistically non-significant decrease of only 3–14% (Figure 9a). Notwithstanding, under C5O5 drying conditions, a notable relative increase of 26% in the total monoterpene content was observed as compared to t_0_ (Figure 9a). This increase was also statistically significant in comparison to all other drying conditions (Figure 9a). Differences of up to 0.38 DW% in monoterpene content were observed across various drying conditions, which could potentially influence the overall aroma of the inflorescence (Figure 9a).

Taken altogether, the sesquiterpene content increased by 15–23% across all tested drying conditions (Figure 9b and Table 2). 

In both the Gen12 and 240 chemovars, it was observed that monoterpenes were more sensitive to the drying process than sesquiterpenes, which could be explained by the higher monoterpene volatility as compared to the sesquiterpenes [35].

Regarding the total terpene content in the Gen12 chemovar, there was an increase during the drying process (ranging from 0.34–0.75 DW%), with C5O5 drying conditions being the only ones statistically significant compared to t_0_ (Figure 9c and Table 2). The higher total terpene content under C5O5 drying conditions was largely attributed to the increase in monoterpene content and the exceptional preservation of β-myrcene in this chemovar (Figure 7 and Figure 9).

The present findings underscore the significant potential of employing drying processes tailored to specific chemovars, in order to optimally preserve the aroma and cannabinoid components as well as reduce mold infestation. In the 240 chemovar, the best drying conditions for both cannabinoid and sesquiterpene preservation were under N_2_ drying conditions, while in the Gen12 chemovar, all drying conditions were suitable for cannabinoid preservation and C5O5 drying conditions were optimal for the overall terpene preservation.

Drying under controlled atmosphere conditions provides a faster alternative to traditional methods while preserving the total terpene content. However, it can modify the concentrations of specific terpenes, potentially impacting the overall aroma profile. The alterations observed in terpene concentrations during the drying process are attributed to degradation, evaporation, and oxidation phenomena [29]. Uziel et al. observed differences in terpene preservation when comparing traditional drying methods to microwave drying at 40 °C [19]. These observations align with our findings and with findings from other drying techniques such as freeze-drying, convection drying, and vacuum-microwave drying [19,28,29]. While hot air drying is effective in retaining CBD content, it can lead to a significant reduction in terpene levels, by 80–90%, particularly at higher temperatures [24]. Moreover, infrared or microwave drying can led to a 75% reduction in total volatile content, specifically due to massive monoterpene degradation during drying [26]. Our study highlights the importance of minimizing drying duration while simultaneously maintaining both cannabinoid and volatile terpene concentrations.

In summary, significant chemical alterations occur in cannabis inflorescences post-harvest, with changes during drying potentially exceeding those in the month before harvest, as suggested by prior studies [36,37]. Given the shorter duration of drying compared to flowering, it is crucial for growers to account for post-harvest changes.

## 3. Materials and Methods

### 3.1. Chemicals

Acetonitrile, anhydrous ammonium formate, ethanol, and formic acid were obtained from Sigma-Aldrich (HPLC grade, Saint Louis, MO, USA). Ultra-pure water was provided by the Milli-Q Plus system (Millipore Corp., Billerica, MA, USA). Cannabinoid analytical standards were purchased from RESTEK (RESTEK, Bellefonte, PA, USA): cannabidivarinic acid (CBDVA), cannabidiolic acid (CBDA), cannabigerolic acid (CBGA), cannabigerol (CBG), cannabidiol (CBD), (−)-Δ9-trans-tetrahydrocannabivarinic acid (THCVA), cannabinol (CBN), (−)-Δ9-trans-tetrahydrocannabinol (Δ-9-THC), (−)-Δ8-trans-tetrahydrocannabinol (Δ-8-THC), (−)-Δ9-trans-tetrahydrocannabinolic acid (THCA), and cannabichromenic acid (CBCA). Each of those standards was obtained at a stock concentration of 1000 µg/mL except CBLA, which was obtained at a stock concentration of 500 µg/mL. Terpene standard mix, at a stock concentration of 2500 µg/mL from each terpene, and which contains the following terpenes—α-pinene, camphene, (−)-β-pinene, β-myrcene, δ-3-carene, α-terpinene, p-cymene, d-limonene, ocimene, γ-terpinene, terpinolene, linalool, (−)-isopulegol, geraniol, β-caryophyllene, α-humulene, nerolidol, (−)-guaiol, and (−)-α-bisabolol—was obtained from RESTEK (RESTEK, Bellefonte, PA, USA).

### 3.2. Plant Material

Fresh medicinal *Cannabis sativa* L. female inflorescences from two different commercially available chemovars—namely, “Gen12” (hybrid chemovar) and “240” (high-THCA chemovar)—were provided by the Barlev farm in May and October 2023, respectively (Bar-Lev Agricultural Crops, Kfar Hess, Israel, 32°15′21.2″ N 34°57′01.0″ E). 

Both chemovars represent prevalent chemovars in Israel (high-THCA and hybrid). Furthermore, both chemovars were accessible to us throughout the duration of the study. Both chemovars were analyzed for their cannabinoid and terpene content at the Agricultural Research Organization, the Department of Food Science, Israel. The cannabis inflorescence, which harbors significant concentrations of cannabinoids and terpenes, is the only part of the plant utilized for both medical and recreational purposes [1,6]. Consequently, it is the focal point of analysis in all pertinent studies within this domain. The sampling method employed in our study aligns with the most widely accepted procedures among researchers and industry practitioners [1,6].

### 3.3. Drying Process

Controlled atmosphere drying chambers were used for the initial drying and curing processes of the fresh cannabis inflorescences. Three different drying environments were examined using the controlled atmosphere chambers—controlled atmospheric conditions (atm), controlled CO_2_ 5%/O_2_ 5%/N_2_ 90% conditions (C5O5), and controlled N_2_ ≥ 99% conditions (N_2_). To attain a relative humidity of less than 10%, we used gases with humidity levels below 0.1%. Additionally, within the controlled atmosphere drying chambers, we inserted 500 g dried silica gel pearls in each chamber (Drying pearls orange, Merck, Darmstadt, Germany) to absorb the moisture during the drying process. The temperature in the drying chambers and open air was set to 15 °C, within the commonly used drying temperature range of 15–21 °C [1]. The controlled atmosphere drying chambers restored the desired drying and curing conditions within the chambers every 30 min. After six days the cannabis inflorescences from both chemovars were completely dry, since no additional inflorescence weight loss was observed (the weight change was within the error range of the water content of 10% ± 1% for the dried inflorescence) and therefore no additional curing step was needed [26]. As a reference system, an open-air drying process was employed for 14 days, followed by one day of curing within a confined breathable tray in the presence of 500 g dried silica gel pearls. This reference system imitated the commonly applied slow industrial drying procedure during the first 14 days of drying. During the drying procedure, all of the cannabis inflorescences were placed in breathable trays within the controlled atmosphere chambers, while the open-air reference samples were placed in an air-ventilated room containing 50–55% relative humidity. In order to determine the water content of the cannabis inflorescence samples, three plates containing weighted cannabis inflorescences from each chemovar were placed in the drying chambers and the weight was recorded before, during, and after drying. Each plate contained three cannabis inflorescences weighing 2–4 g. For calculating the cannabinoid and terpene content, the fresh cannabis inflorescences weight at the initial time point (t_0_) was normalized to dry weight using the average weight loss for each chemovar.

### 3.4. Sample Preparation

Fresh or dried cannabis inflorescences from the 240 and Gen12 chemovars were ground homogenously with a mortar and pestle in the presence of liquid nitrogen, providing 5 replicates from each treatment group per chemovar. The homogenously ground cannabis samples (500 ± 0.5 mg for fresh inflorescences and 100 ± 0.1 mg for dried inflorescences) were extracted with 4 mL of ethanol in 15 mL Falcon tubes and shaken (Digital Orbital Shaker, MRC, Holon, Israel) in the dark for 15 min at 500 rpm. One mL of the extract was transferred to an Eppendorf tube and centrifuged for 4 min at 12,000 rpm. For the determination of cannabinoid levels, a dilution of 1:11 of the supernatant with ethanol was carried out, and 1 mL aliquot was transferred to an HPLC vial and subjected to high-pressure liquid chromatography–photodiode arrays (HPLC–PDA) analysis. For the determination of terpene levels, 0.25 mL of the supernatant was inserted into a GC vial and analyzed via gas chromatography–mass spectroscopy (GC/MS).

### 3.5. Quantification of Cannabinoids by HPLC–PDA and Terpenes by GC/MS

The ethanolic cannabis extracts were analyzed as described in Birenboim et al., utilizing HPLC–PDA (Acquity Arc FTN-R; Model PDA-2998, Waters Corp., Milford, MA, USA) equipped with Kinetex^®^ 1.7 μm XB-C18 100 A LC column (150 × 2.1 mm i.d. and 1.7 μm particle size; Phenomenex, Torrance, CA, USA) for the cannabinoids analysis [18]. The cannabinoids were quantified by comparing the integrated peak area with the corresponding cannabinoid calibration curve ranging from 1 to 1000 µg/mL (Appendix A) [18].

The terpene analysis was carried out by GC/MS (Agilent, Santa Clara, CA, USA) as recently reported by Birenboim et al. utilizing a DB-5 capillary column (5% phenyl, 95% dimethylpolysiloxane, 30 m × 0.250 mm, 0.25 m; Agilent, Santa Clara, CA, USA) for analyte separation [18]. The terpenes were quantified by comparing the integrated peak area with the corresponding terpene calibration curve ranging from 0.5 to 250 µg/mL (Appendix A) [18]. The methods’ analytical validation parameters (i.e., R^2^, limit of detection, limit of quantification, repeatability, and accuracy) were recently published in Birenboim et al. [18].

### 3.6. Microbiological Assay

Wet inflorescence at t_0_, dry inflorescence from CA drying chambers at day 6, and wet open-air inflorescence at day 12 of the 240 chemovar were tested for microbiological contamination using the colony forming unit (CFU) of total yeasts and molds and real-time PCR. For total yeasts and molds analysis, fresh or dried cannabis inflorescences from the 240 chemovar were ground homogenously with a mortar and pestle in the presence of liquid nitrogen, providing three replicates from each sample. For the CFU test, 50 ± 0.1 mg of the homogenously ground cannabis samples were inserted into 1 mL of sterilized distilled water solution in a 1.5 mL Eppendorf tube and vortexed for 10–15 s. Thereafter, 100, 1000, and 10,000 times dilutions of the solutions were prepared. Subsequently, 100 µL of the diluted solutions were spread on Potato Dextrose Agar (PDA) plates supplemented with 0.025% chloramphenicol and incubated at room temperature for 4 days to develop yeasts and molds. The microbial count of each plate was then reported as the CFU per dry gram of each sample (CFU/g dry).

The relative fungal biomass was determined as described previously in Li et al. [38]. Briefly, each cannabis inflorescence was ground homogenously with a mortar and pestle in the presence of liquid nitrogen. From each inflorescence, three samples of approximately 50 mg were taken for DNA extraction using Wizard^®^ genomic DNA purification kits (Promega, Madison, WI, USA) according to the manufacturer’s instructions. DNA quantity and quality were determined by the NanoDrop One (Thermo Fisher Scientific, Waltham, MA, USA) spectrophotometer. The extracted DNA was diluted for 10 ng/µL and 1 ng/µL for further analysis. The relative biomass of *Botrytis cinerea* and *Alternaria alternata* was evaluated by an RT-qPCR analysis conducted with a Step One Plus Real-Time PCR (Applied Biosystems, Waltham, MA, USA). PCR amplification was performed with 2.5 µL of a diluted DNA template (10 ng/µL) in a 10 µL reaction mixture containing 5 µL Syber Green (Applied Biosystems, Waltham, MA, USA) and 250 nM primers. The qRT-PCR analysis was conducted with the corresponding primer sets of the selected fungi: forward, 5′-TGCTCCAGAAGCTTTGTTCCAA-3′, and reverse, 5′-TCGGAGATACCTGGGTACATAG-3′, for the *B. cinerea* actin gene, and forward, 5′-TTGGACTGCTCTAGCCTGGT-3′, and reverse, 5′-GTCAAACACGTGCGATAACC-3′, for the *A. alternata* actin gene. The PCR cycling program included: 10 min at 94 °C, followed by 40 cycles at 94 °C for 10 s, 60 °C for 15 s, and 72 °C for 20 s. The relative biomass of the selected genes was normalized using Ct values of the Cannabis 18S rRNA gene (forward, 5′-TTCGTCCTCCCCCAAAAGT-3′, and reverse, 5′-CCGAGCGTTTTGTTCTTTCG-3′) as the reference gene, and expression values were calculated relatively to the uninfected sample using Step One software v2.2.2 (Applied Biosystems, Waltham, MA, USA). Each treatment consisted of three biological repeats and three technical replicates.

### 3.7. Statistical Analysis and Cannabinoid/Terpene Content Calculations

For each compound analyzed, one-way ANOVA followed by Tukey’s post hoc test was used to determine the differences in cannabinoid and terpene concentrations between the different drying conditions and t_0_, for both chemovars, at α = 0.05 using GraphPad PRISM 10 (San Diego, CA, USA).

Relative concentration was defined as the dry concentration at any specified time (in DW%) divided by the concentration at t_0_ (normalized to DW%) and was calculated for each compound as described in Equation (1).
(1)$$Relative\;concentration=\frac{DW\%\;after\;drying\;}{normalized\;DW\%\;at\;t_{0}}$$

Total THC and total CBD content were calculated according to Equations (2) and (3), respectively [24].
(2)$$total\;THC\;[DW\%]=\left(THCA\;DW\%\right)*0.877+THC\;DW\%$$
(3)$$total\;CBD\;[DW\%]=\left(CBDA\;DW\%\right)*0.877+CBD\;DW\%$$

## 4. Conclusions

This study explored the impact of varying fast drying conditions on the chemical composition of cannabis inflorescences. The drying environment notably affected the chemical composition of cannabis inflorescences. Compared to traditional methods, controlled atmosphere chambers reduced the drying and curing time by at least 60%, without reducing the total volatile terpene content and without encouraging mold growth. On the other hand, inflorescences from the 240 chemovar subjected to traditional drying condition were highly infested by *Alternaria alternata* and also revealed low infestation of *Botrytis cinerea*; consequently, they are less ideal for routine commercial use. The different drying conditions employed in the present study affected the cannabinoid composition in both chemovars differently. In the 240 chemovar, controlled N_2_ and atm drying conditions were able to preserve THCA levels comparable to t_0_, while in the Gen12 chemovar all of the employed drying conditions preserved THCA and CBDA content. On the other hand, in both chemovars, open-air drying conditions resulted in a larger extent of decarboxylation of the major cannabinoids as compared to the controlled atmosphere drying conditions, resulting in 3–8-fold higher THC and CBD concentrations compared to t_0_. A decrease in CBGA concentration was observed in both chemovars, and the lowest CBGA concentration after the drying process was observed under open-air drying conditions.

Regarding the aroma components, C5O5 drying conditions were optimal for preserving monoterpenes. On the other hand, N_2_ drying conditions were the only drying conditions yielding a statistically significant increase in sesquiterpene content in both chemovars.

## Figures and Tables

**Figure 1 plants-13-01049-f001:**
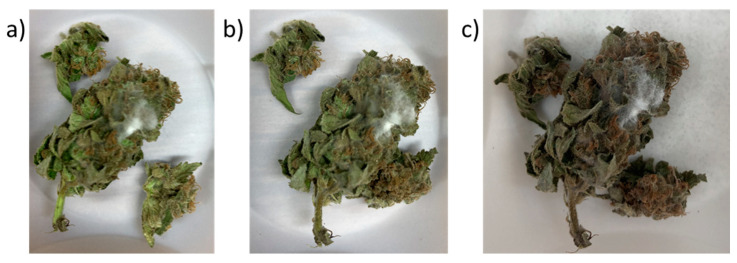
Mycelium development in the 240 inflorescence: (**a**) at day 12 of drying, (**b**) at day 14 and (**c**) at day 15 after one day of curing in the presence of dried silica gel pearls.

**Figure 2 plants-13-01049-f002:**
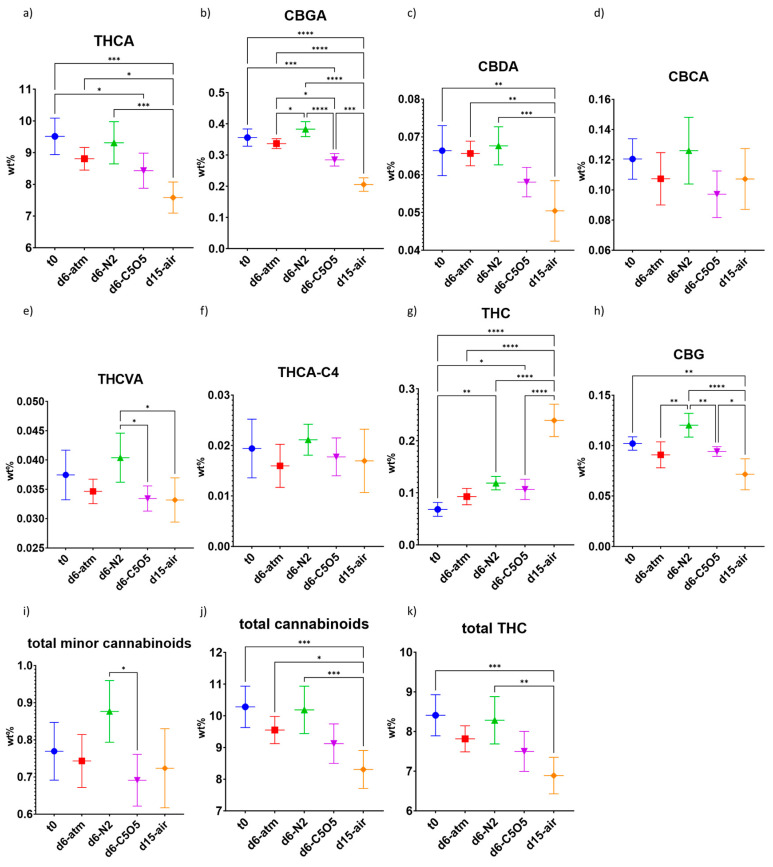
Mean cannabinoid concentrations (in DW%, *y*-axis, for each drying procedure and t_0_; n = 5) of the 240 chemovar determined under four different drying conditions by HPLC–PDA: controlled atmospheric conditions (atm), controlled N_2_ ≥ 99% conditions (N_2_), controlled CO_2_ 5%/O_2_ 5%/N_2_ 90% conditions (C5O5), and open-air drying process used as a reference system (air). (**a**) THCA, (**b**) CBGA, (**c**) CBDA, (**d**) CBCA, (**e**) THCVA, (**f**) THCA-C4, (**g**) THC, (**h**) CBG, (**i**) total minor cannabinoids, (**j**) total cannabinoids and (**k**) total THC. Statistical significance between the different drying conditions and t_0_ for each compound was calculated using one-way ANOVA followed by Tukey’s post hoc test (*p* value < 0.0332 (*), *p* < 0.0021 (**), *p* < 0.0002 (***), *p* < 0.0001 (****)).

**Figure 3 plants-13-01049-f003:**
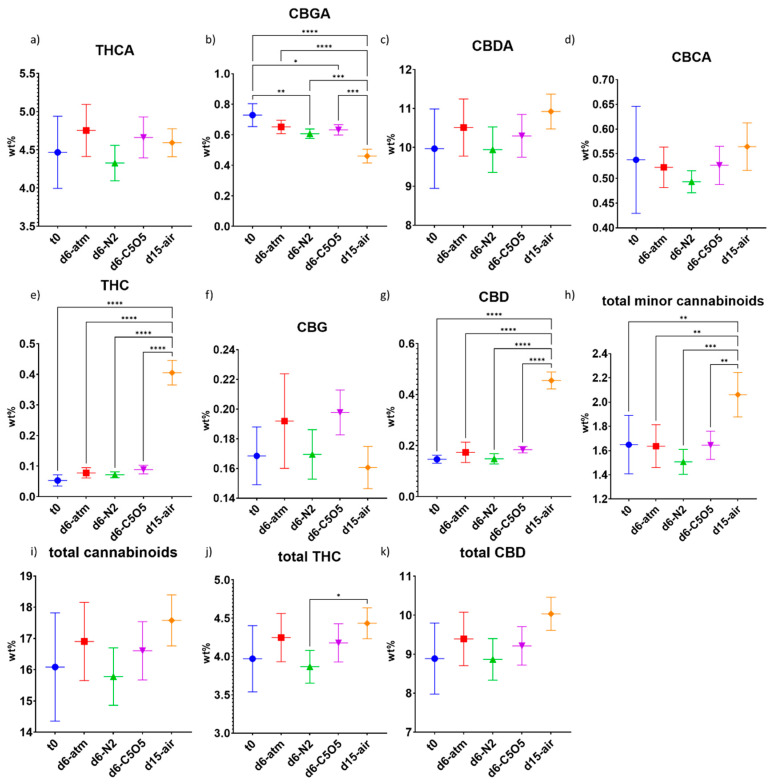
Mean cannabinoid concentrations (in DW%, *y*-axis, for each drying procedure and t_0_; n = 5) of the Gen12 chemovar determined under four different drying conditions by HPLC–PDA: controlled atmospheric conditions (atm), controlled N_2_ ≥ 99% conditions (N_2_), controlled CO_2_ 5%/O_2_ 5%/N_2_ 90% conditions (C5O5), and open-air drying process used as a reference system (air). (**a**) THCA, (**b**) CBGA, (**c**) CBDA, (**d**) CBCA, (**e**) THC, (**f**) CBG, (**g**) CBD, (**h**) total minor cannabinoids, (**i**) total cannabinoids, (**j**) total THC and (**k**) total CBD. Statistical significance between the different drying conditions and t_0_ for each compound was calculated using one-way ANOVA followed by Tukey’s post hoc test (*p* value < 0.0332 (*), *p* < 0.0021 (**), *p* < 0.0002 (***), *p* < 0.0001 (****)).

**Figure 4 plants-13-01049-f004:**
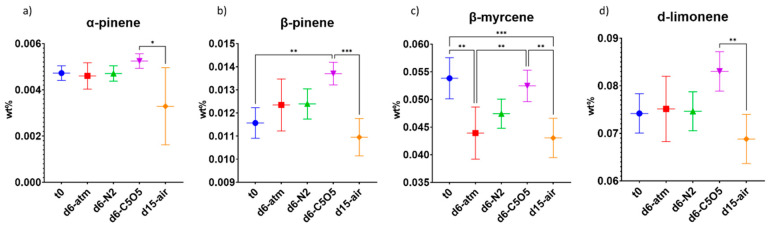
Mean monoterpene concentrations (in DW%, *y*-axis, for each drying procedure and t_0_; n = 5) of the 240 chemovar determined under four different drying conditions by GC/MS: controlled atmospheric conditions (atm), controlled N_2_ ≥ 99% conditions (N_2_), controlled CO_2_ 5%/O_2_ 5%/N_2_ 90% conditions (C5O5), and open-air drying process used as a reference system (air). (**a**) α-pinene, (**b**) (−)-β-pinene, (**c**) β-myrcene and (**d**) d-limonene. Statistical significance between the different drying conditions and t_0_ for each compound was calculated using one-way ANOVA followed by Tukey’s post hoc test (*p* value < 0.0332 (*), *p* < 0.0021 (**), *p* < 0.0002 (***)).

**Figure 5 plants-13-01049-f005:**
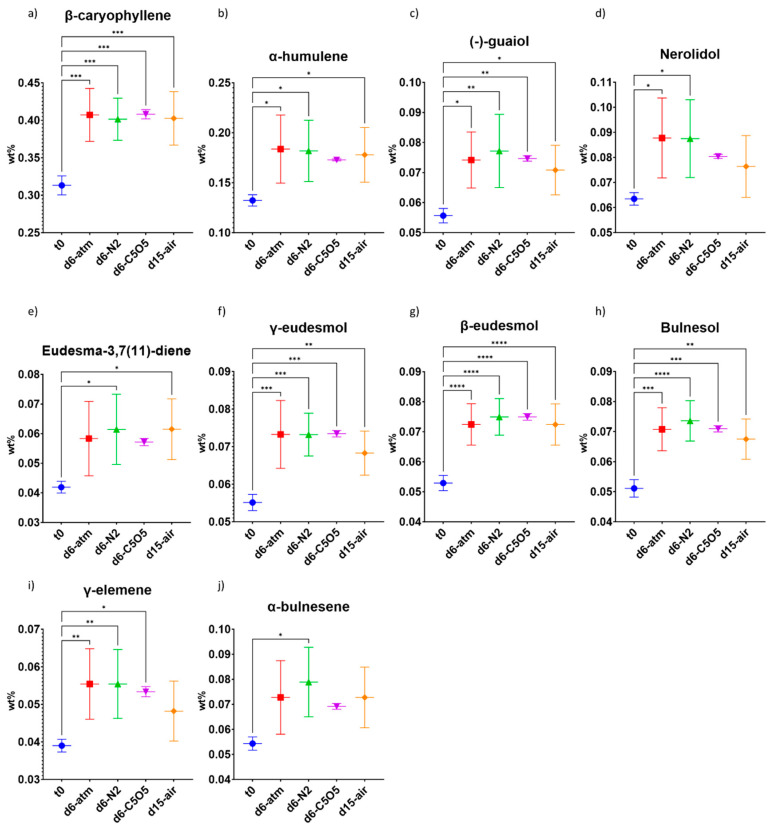
Mean sesquiterpene concentrations (in DW%, *y*-axis, for each drying procedure and t_0_; n = 5) of the 240 chemovar determined under four different drying conditions by GC/MS: controlled atmospheric conditions (atm), controlled N_2_ ≥ 99% conditions (N_2_), controlled CO_2_ 5%/O_2_ 5%/N_2_ 90% conditions (C5O5), and open-air drying process used as a reference system (air). (**a**) β-caryophyllene, (**b**) α-humulene, (**c**) (−)-guaiol, (**d**) nerolidol, (**e**) eudesma-3,7(11)-diene, (**f**) γ-eudesmol, (**g**) β-eudesmol, (**h**) bulnesol, (**i**) γ-elemene and (**j**) α-bulnesene. Statistical significance between the different drying conditions and t_0_ for each compound was calculated using one-way ANOVA followed by Tukey’s post hoc test (*p* value < 0.0332 (*), *p* < 0.0021 (**), *p* < 0.0002 (***), *p* < 0.0001 (****)).

**Figure 6 plants-13-01049-f006:**
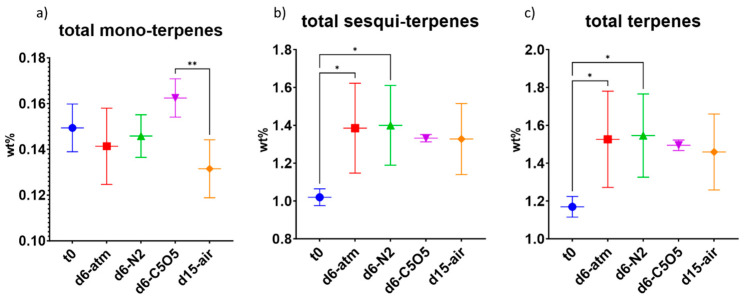
Mean total terpene content (in DW%, *y*-axis, for each drying procedure and t_0_; n = 5) of the 240 chemovar determined under four different drying conditions by GC/MS: controlled atmospheric conditions (atm), controlled N_2_ ≥ 99% conditions (N_2_), controlled CO_2_ 5%/O_2_ 5%/N_2_ 90% conditions (C5O5), and open-air drying process used as a reference system (air). (**a**) total monoterpenes, (**b**) total sesquiterpenes and (**c**) total terpenes. Statistical significance between the different drying conditions and t_0_ for each compound was calculated using one-way ANOVA followed by Tukey’s post hoc test (*p* value < 0.0332 (*), *p* < 0.0021 (**)).

**Figure 7 plants-13-01049-f007:**
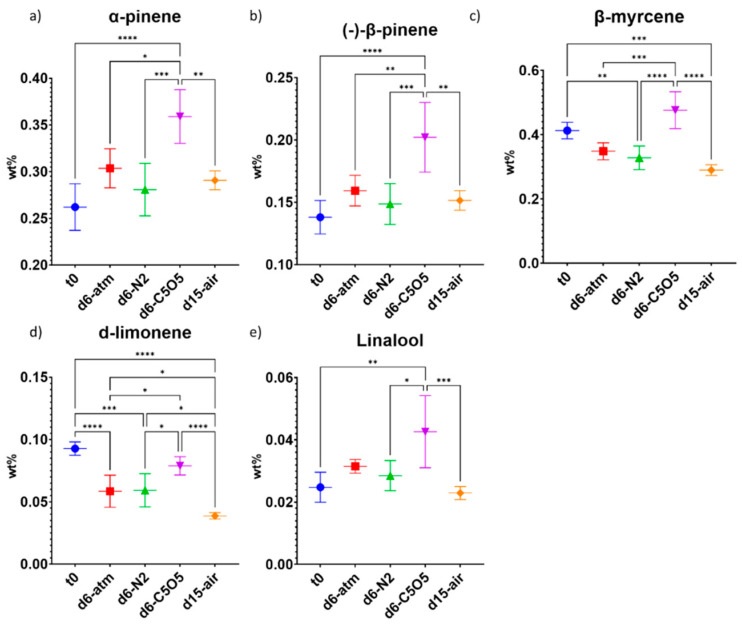
Mean monoterpene concentrations (in DW%, *y*-axis, for each drying procedure and t_0_; n = 5) of the Gen12 chemovar determined under four different drying conditions by GC/MS: controlled atmospheric conditions (atm), controlled N_2_ ≥ 99% conditions (N_2_), controlled CO_2_ 5%/O_2_ 5%/N_2_ 90% conditions (C5O5), and open-air drying process used as a reference system (air). (**a**) α-pinene, (**b**) (−)-β-pinene, (**c**) β-myrcene, (**d**) d-limonene and (**e**) linalool. Statistical significance between the different drying conditions and t_0_ for each compound was calculated using one-way ANOVA followed by Tukey’s post hoc test (*p* value < 0.0332 (*), *p* < 0.0021 (**), *p* < 0.0002 (***), *p* < 0.0001 (****)).

**Figure 8 plants-13-01049-f008:**
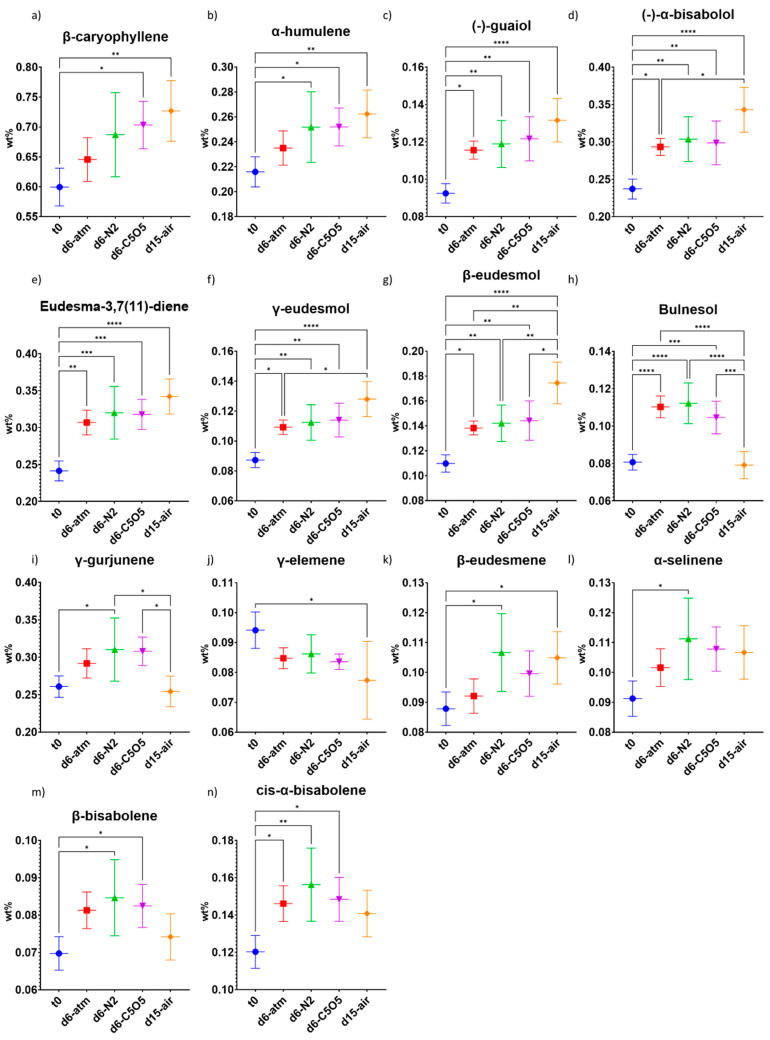
Mean sesquiterpene concentrations (in DW%, *y*-axis, for each drying procedure and t_0_; n = 5) of the Gen12 chemovar measured under the four different drying conditions by GC/MS: controlled atmospheric conditions (atm), controlled N_2_ ≥ 99% conditions (N_2_), controlled CO_2_ 5%/O_2_ 5%/N_2_ 90% conditions (C5O5), and open-air drying process used as a reference system (air). (**a**) β-caryophyllene, (**b**) α-humulene, (**c**) (−)-guaiol, (**d**) (−)-α-bisabolol, (**e**) eudesma-3,7(11)-diene, (**f**) γ-eudesmol, (**g**) β-eudesmol, (**h**) bulnesol, (**i**) γ-gurjunene, (**j**) γ-elemene, (**k**) β-eudesmene, (**l**) α-selinene, (**m**) β-bisabolene and (**n**) cis-α-bisabolene. Statistical significance between the different drying conditions and t_0_ for each compound was calculated using one-way ANOVA followed by Tukey’s post hoc test (*p* value < 0.0332 (*), *p* < 0.0021 (**), *p* < 0.0002 (***), *p* < 0.0001 (****)).

**Figure 9 plants-13-01049-f009:**
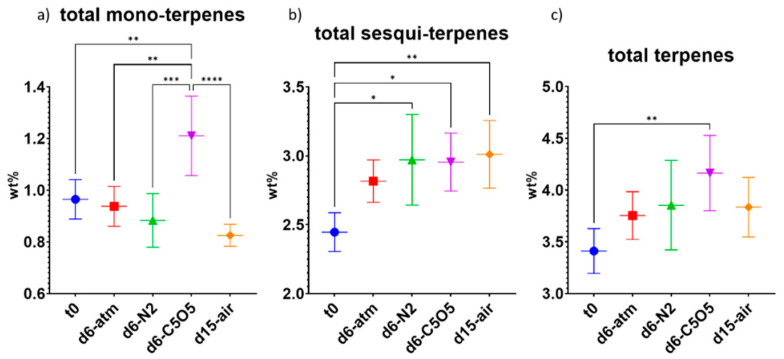
Mean total terpene content (in DW%, *y*-axis, for each drying procedure and t_0_; n = 5) of the Gen12 chemovar measured under the four different drying conditions by GC/MS: controlled atmospheric conditions (atm), controlled N_2_ ≥ 99% conditions (N_2_), controlled CO_2_ 5%/O_2_ 5%/N_2_ 90% conditions (C5O5), and open-air drying process used as a reference system (air). (**a**) total monoterpenes, (**b**) total sesquiterpenes and (**c**) total terpenes. Statistical significance between the different drying conditions and t_0_ for each compound was calculated using one-way ANOVA followed by Tukey’s post hoc test (*p* value < 0.0332 (*), *p* < 0.0021 (**), *p* < 0.0002 (***), *p* < 0.0001 (****)).

**Table 1 plants-13-01049-t001:** The 240 and Gen12 cannabinoid composition at harvest day (t_0_, absolute concentration, normalized to DW%) and absolute cannabinoid concentrations (in DW%) after drying under four drying conditions. For each drying procedure and t_0_ n = 5.

Cannabinoid	Absolute Concentration ± SE at t_0_	Absolute Concentration ± SE after 6 Days of Controlled Atmospheric Drying			Absolute Concentration ± SE after 15 Days of Drying and Curing (Traditional Drying)
240					
		Atm	N_2_	C5O5	Open-air
CBDVA	<LOD				
CBDA	0.066 ± 0.007	0.066 ± 0.003 (ns) ^a^	0.067 ± 0.005 (ns)	0.058 ± 0.004 (ns)	0.050 ± 0.008 (**) ^b^
CBGA	0.36 ± 0.03	0.34 ± 0.02 (ns)	0.38 ± 0.02 (ns)	0.28 ± 0.02 (***)	0.20 ± 0.02 (****)
CBG	0.102 ± 0.007	0.09 ± 0.01 (ns)	0.12 ± 0.01 (ns)	0.094 ± 0.005 (ns)	0.07 ± 0.02 (**)
CBD	<LOD				
THCVA	0.037 ± 0.004	0.035 ± 0.002 (ns)	0.040 ± 0.004 (ns)	0.033 ± 0.002 (ns)	0.033 ± 0.004 (ns)
THCA-C4	0.020 ± 0.006	0.016 ± 0.004 (ns)	0.021 ± 0.003 (ns)	0.018 ± 0.004 (ns)	0.017 ± 0.006 (ns)
THC	0.07 ± 0.01	0.09 ± 0.02 (ns)	0.12 ± 0.01 (**)	0.10 ± 0.02 (*)	0.24 ± 0.03 (****)
THCA	9.5 ± 0.7	8.8 ± 0.4 (ns)	9.3 ± 0.7 (ns)	8.4 ± 0.6 (*)	7.6 ± 0.5 (***)
CBCA	0.12 ± 0.01	0.11 ± 0.02 (ns)	0.13 ± 0.02 (ns)	0.10 ± 0.02 (ns)	0.11 ± 0.02 (ns)
Total cannabinoids	10.3 ± 0.6	9.5 ± 0.47 (ns)	10.2 ± 0.78 (ns)	9.1 ± 0.67 (ns)	8.3 ± 0.6 (***)
Total minor cannabinoids	0.80 ± 0.08	0.74 ± 0.07 (ns)	0.90 ± 0.08 (ns)	0.70 ± 0.07 (ns)	0.7 ± 0.1 (ns)
total THC	8.4 ± 0.5	7.8 ± 0.3 (ns)	8.3 ± 0.6 (ns)	7.5 ± 0.5 (ns)	6.9 ± 0.5 (***)
Gen12					
		Atm	N_2_	C5O5	Open-air
CBDVA	0.015 ± 0.004	0.020 ± 0.003 (ns)	0.018 ± 0.004 (ns)	0.016 ± 0.003 (ns)	0.016 ± 0.003 (ns)
CBDA	10.0 ± 1.0	10.5 ± 0.7 (ns)	10.0 ± 0.6 (ns)	10.3 ± 0.5 (ns)	10.9 ± 0.4 (ns)
CBGA	0.73 ± 0.08	0.65 ± 0.04 (ns)	0.61 ± 0.03 (**)	0.63 ± 0.03 (*)	0.46 ± 0.04 (****)
CBG	0.17 ± 0.02	0.19 ± 0.03 (ns)	0.17 ± 0.02 (ns)	0.20 ± 0.02 (ns)	0.16 ± 0.01 (ns)
CBD	0.15 ± 0.02	0.17 ± 0.04 (ns)	0.15 ± 0.02 (ns)	0.18 ± 0.01 (ns)	0.45 ± 0.03 (****)
THCVA	<LOD				
THCA-C4	<LOD				
THC	0.05 ± 0.02	0.08 ± 0.02 (ns)	0.07 ± 0.01 (ns)	0.09 ± 0.01 (ns)	0.40 ± 0.04 (****)
THCA	4.5 ± 0.5	4.8 ± 0.3 (ns)	4.3 ± 0.2 (ns)	4.7 ± 0.3 (ns)	4.6 ± 0.2 (ns)
CBCA	0.54 ± 0.11	0.52 ± 0.04 (ns)	0.49 ± 0.02 (ns)	0.53 ± 0.04 (ns)	0.56 ± 0.05 (ns)
Total cannabinoids	16.1 ± 1.7	16.9 ± 1.2 (ns)	15.8 ± 0.9 (ns)	16.6 ± 0.9 (ns)	17.6 ± 0.8 (ns)
Total minor cannabinoids	1.6 ± 0.2	1.6 ± 0.2 (ns)	1.5 ± 0.1 (ns)	1.6 ± 0.1 (ns)	2.1 ± 0.2 (**)
Total THC	4.0 ± 0.4	4.2 ± 0.3 (ns)	3.9 ± 0. 2 (ns)	4.2 ± 0.2 (ns)	4.4 ± 0.2 (ns)
Total CBD	8.9 ± 0.9	9.4 ± 0.7 (ns)	8.9 ± 0.5 (ns)	9.2 ± 0.5 (ns)	10.0 ± 0.4 (ns)

^a^ ns, not significant; ^b^ One-way ANOVA results of absolute cannabinoid concentrations between the different drying conditions and t_0_ at an adjusted significance value of *p* < 0.05. *p* value < 0.0332 (*), *p* < 0.0021 (**), *p* < 0.0002 (***), *p* < 0.0001 (****).

**Table 2 plants-13-01049-t002:** The 240 and Gen12 terpene composition at harvest day (t_0_, absolute concentration, normalized to DW%) and absolute terpene concentrations (in DW%) after drying under the four drying conditions. For each drying procedure and t_0_ n = 5.

Terpene	Absolute Concentration ± SE at t_0_	Absolute Concentration ± SE after 6 Days of Controlled Atmospheric Drying			Absolute Concentration ± SE after 15 Days of Drying and Curing (Traditional Drying)
240					
		Atm	N_2_	C5O5	Open-air
α-pinene	0.0047 ± 0.0003	0.0046 ± 0.0006 ^a^	0.0047 ± 0.0003	0.0052 ± 0.0003	0.003 ± 0.002
Camphene	0.0020 ± 0.0001	0.001 ± 0.001	0.0018 ± 0.0009	0.0024 ± 0.0001	0.0004 ± 0.0004
(−)-β-pinene	0.0116 ± 0.0007	0.012 ± 0.001	0.0124 ± 0.0006	0.0137 ± 0.0005	0.0110 ± 0.0008
β-myrcene	0.054 ± 0.004	0.044 ± 0.005 (**) ^b^	0.047 ± 0.003 (ns) ^c^	0.052 ± 0.003 (ns)	0.043 ± 0.004 (***)
δ-3-carene	<LOD				
d-limonene	0.074 ± 0.004	0.075 ± 0.007 (ns)	0.075 ± 0.004 (ns)	0.083 ± 0.004 (ns)	0.069 ± 0.005 (ns)
Linalool	<LOD				
Fenchol	0.003 ± 0.002	0.004 ± 0.002	0.0048 ± 0.0007	0.0057 ± 0.0005	0.0050 ± 0.0006
Pinalol	<LOD				
β-caryophyllene	0.31 ± 0.01	0.41 ± 0.04 (***)	0.40 ± 0.03 (***)	0.408 ± 0.006 (***)	0.40 ± 0.04 (***)
α-humulene	0.132 ± 0.006	0.18 ± 0.03 (*)	0.18 ± 0.03 (*)	0.173 ± 0.001 (ns)	0.18 ± 0.03 (*)
(−)-guaiol	0.056 ± 0.002	0.074 ± 0.009 (*)	0.08 ± 0.01 (**)	0.0747 ± 0.0009 (**)	0.071 ± 0.008 (*)
(−)-α-bisabolol	<LOD				
Nerolidol	0.063 ± 0.002	0.09 ± 0.02 (*)	0.09 ± 0.02 (*)	0.0804 ± 0.0008 (ns)	0.08 ± 0.01 (ns)
γ-elemene	0.039 ± 0.002	0.055 ± 0.009 (**)	0.055 ± 0.009 (**)	0.053 ± 0.001 (*)	0.048 ± 0.008 (ns)
α-bergomotene	0.0152 ± 0.0006	0.020 ± 0.006	0.022 ± 0.006	0.0183 ± 0.0002	0.021 ± 0.005
α-guaiene	0.028 ± 0.001	0.04 ± 0.01	0.04 ± 0.01	0.0350 ± 0.0005	0.04 ± 0.01
β-farensene	0.023 ± 0.001	0.03 ± 0.02	0.03 ± 0.01	0.024 ± 0.001	0.029 ± 0.009
β-eudesmene	0.0124 ± 0.0006	0.015 ± 0.001	0.0166 ± 0.0005	0.0162 ± 0.0006	0.016 ± 0.001
α-selinene	0.0164 ± 0.0008	0.024 ± 0.007	0.026 ± 0.007	0.0220 ± 0.0005	0.024 ± 0.0006
α-bulnesene	0.054 ± 0.003	0.07 ± 0.01 (ns)	0.08 ± 0.01 (*)	0.070 ± 0.001 (ns)	0.07 ± 0.01 (ns)
β-bisabolene	<LOD				
cis-α-bisabolene	<LOD				
Eudesma-3,7(11)-diene	0.042 ± 0.002	0.06 ± 0.01 (ns)	0.06 ± 0.01 (*)	0.057 ± 0.001 (ns)	0.06 ± 0.01 (*)
γ-eudesmol	0.055 ± 0.002	0.073 ± 0.009 (***)	0.073 ± 0.005 (***)	0.0734 ± 0.0009 (***)	0.068 ± 0.006 (**)
β-eudesmol	0.053 ± 0.002	0.072 ± 0.007 (****)	0.075 ± 0.006 (****)	0.075 ± 0.001 (****)	0.072 ± 0.007 (****)
Bulnesol	0.051 ± 0.003	0.071 ± 0.007 (***)	0.074 ± 0.007 (****)	0.071 ± 0.001 (***)	0.067 ± 0.007 (**)
α-gurjunene	0.0136 ± 0.0005	0.02 ± 0.01	0.022 ± 0.009	0.0175 ± 0.0002	0.021 ± 0.007
γ-gurjunene	0.031 ± 0.001	0.04 ± 0.01	0.04 ± 0.01	0.0403 ± 0.0005	0.039 ± 0.008
Elemol	0.0162 ± 0.0006	0.020 ± 0.006	0.020 ± 0.006	0.0172 ± 0.0001	0.012 ± 0.003
Total terpenes	1.17 ± 0.05	1.55 ± 0.22 (*)	1.55 ± 0.21 (*)	1.49 ± 0.03 (ns)	1.43 ± 0.21 (ns)
Total monoterpenes	0.15 ± 0.01	0.15 ± 0.02 (ns)	0.146 ± 0.009 (ns)	0.162 ± 0.008 (ns)	0.13 ± 0.01 (ns)
Total sesquiterpenes	1.02 ± 0.04	1.4 ± 0.2 (*)	1.4 ± 0.2 (*)	1.33 ± 0.02 (ns)	1.3 ± 0.2 (ns)
Gen12					
		Atm	N_2_	C5O5	Open-air
α-pinene	0.26 ± 0.02	0.30 ± 0.02 (ns)	0.28 ± 0.03 (ns)	0.36 ± 0.03 (****)	0.29 ± 0.01 (ns)
Camphene	0.005 ± 0.001	0.0057 ± 0.0005	0.0054 ± 0.0006	0.0069 ± 0.0004	0.0053 ± 0.0003
(−)-β-pinene	0.14 ± 0.01	0.16 ± 0.01 (ns)	0.15 ± 0.02 (ns)	0.20 ± 0.03 (****)	0.151 ± 0.008 (ns)
β-myrcene	0.41 ± 0.03	0.35 ± 0.03 (ns)	0.33 ± 0.04 (**)	0.48 ± 0.06 (ns)	0.29 ± 0.02 (***)
δ-3-carene	<LOD				
d-limonene	0.093 ± 0.005	0.06 ± 0.01 (****)	0.06 ± 0.01 (***)	0.079 ± 0.007 (ns)	0.039 ± 0.002 (****)
Linalool	0.025 ± 0.005	0.032 ± 0.002 (ns)	0.028 ± 0.005 (ns)	0.04 ± 0.01 (**)	0.023 ± 0.002 (ns)
Fenchol	0.011 ± 0.001	0.012 ± 0.001	0.013 ± 0.002	0.016 ± 0.004	0.010 ± 0.002
Pinalol	0.018 ± 0.001	0.0192 ± 0.0008	0.020 ± 0.002	0.024 ± 0.006	0.017 ± 0.001
β-caryophyllene	0.60 ± 0.03	0.64 ± 0.04 (ns)	0.69 ± 0.07 (ns)	0.70 ± 0.04 (*)	0.73 ± 0.05 (**)
α-humulene	0.22 ± 0.01	0.24 ± 0.01 (ns)	0.25 ± 0.03 (*)	0.25 ± 0.02 (*)	0.26 ± 0.02 (**)
(−)-guaiol	0.092 ± 0.005	0.116 ± 0.005 (*)	0.12 ± 0.01 (**)	0.12 ± 0.01 (**)	0.13 ± 0.01 (****)
(−)-α-bisabolol	0.24 ± 0.01	0.29 ± 0.01 (*)	0.30 ± 0.03 (**)	0.30 ± 0.03 (**)	0.34 ± 0.03 (****)
Nerolidol	<LOD				
γ-elemene	0.094 ± 0.006	0.085 ± 0.003 (ns)	0.086 ± 0.006 (ns)	0.084 ± 0.003 (ns)	0.08 ± 0.01 (*)
α-bergomotene	0.032 ± 0.002	0.035 ± 0.002	0.036 ± 0.005	0.038 ± 0.002	0.036 ± 0.002
α-guaiene	<LOD				
β-farensene	0.026 ± 0.002	0.030 ± 0.003	0.032 ± 0.004	0.031 ± 0.002	0.028 ± 0.003
β-eudesmene	0.088 ± 0.006	0.092 ± 0.006 (ns)	0.11 ± 0.01 (*)	0.100 ± 0.008 (ns)	0.105 ± 0.009 (*)
α-selinene	0.091 ± 0.006	0.102 ± 0.006 (ns)	0.11 ± 0.01 (*)	0.108 ± 0.007 (ns)	0.107 ± 0.009 (ns)
α-bulnesene	<LOD				
β-bisabolene	0.070 ± 0.004	0.081 ± 0.005 (ns)	0.08 ± 0.01 (*)	0.082 ± 0.006 (*)	0.074 ± 0.006 (ns)
cis-α-bisabolene	0.120 ± 0.009	0.15 ± 0.01 (*)	0.16 ± 0.02 (**)	0.15 ± 0.01 (*)	0.14 ± 0.01 (ns)
Eudesma-3,7(11)-diene	0.24 ± 0.01	0.31 ± 0.02 (**)	0.32 ± 0.04 (***)	0.32 ± 0.02 (***)	0.34 ± 0.02 (****)
γ-eudesmol	0.087 ± 0.005	0.109 ± 0.005 (*)	0.11 ± 0.01 (**)	0.11 ± 0.01 (**)	0.13 ± 0.01 (****)
β-eudesmol	0.110 ± 0.007	0.138 ± 0.006 (*)	0.14 ± 0.01 (**)	0.14 ± 0.02 (**)	0.17 ± 0.02 (****)
Bulnesol	0.081 ± 0.004	0.110 ± 0.006 (****)	0.11 ± 0.01 (****)	0.104 ± 0.009 (***)	0.079 ± 0.007 (ns)
α-gurjunene	<LOD				
γ-gurjunene	0.26 ± 0.01	0.29 ± 0.02 (ns)	0.31 ± 0.04 (*)	0.31 ± 0.02 (ns)	0.25 ± 0.02 (ns)
Elemol	<LOD				
Total terpenes	3.41 ± 0.22	3.75 ± 0.23 (ns)	3.85 ± 0.4 (ns)	4.15 ± 0.35 (**)	3.8 ± 0.24 (ns)
Total monoterpenes	0.96 ± 0.08	0.94 ± 0.08 (ns)	0.85 ± 0.10 (ns)	1.2 ± 0.15 (**)	0.80 ± 0.04 (ns)
Total sesquiterpenes	2.45 ± 0.14	2.81 ± 0.15 (ns)	3.0 ± 0.3 (*)	3.0 ± 0.2 (*)	3.0 ± 0.2 (**)

^a^ Statistical analysis conducted only for terpene with initial concentrations higher than 0.04 DW%. ^b^ One-way ANOVA results of absolute terpene concentrations between the different drying conditions and t_0_ at a significance value of *p* < 0.05. *p* value < 0.0332 (*), *p* < 0.0021 (**), *p* < 0.0002 (***), *p* < 0.0001 (****). ^c^ ns, not significant.

**Table 3 plants-13-01049-t003:** Percentage weight loss during the drying of the 240 and Gen12 chemovars under controlled atmosphere and open-air drying conditions.

Drying Conditions	240	Gen12
Weight loss in controlled atmosphere drying chambers after 6 days	78.5±1.6%	77.4±2.5%
Weight loss in open-air drying conditions after 14 days (before curing)	60.5±2.4% (**) ^a^	73±1.5% (*) ^a^
Weight loss in open-air drying conditions after 15 days (including curing)	75.8±1.5% (ns) ^a^	80±4% (ns) ^a^

^a^ *t*-test statistical results comparing between controlled atmosphere and open-air drying conditions (*p* value < 0.0332 (*); *p* < 0.0021 (**); ns, not significant).

**Table 4 plants-13-01049-t004:** Mold determination of the 240 chemovar inflorescences at t_0_, under controlled atmosphere drying conditions and open-air drying conditions.

Drying Conditions	CFU/g Dry × 10^6^	*Alternaria alternata* Extent of Infestation [2^−ΔΔCt^]	*Botrytis cinerea* Extent of Infestation [2^−ΔΔCt^]
Fresh inflorescences at t_0_	2.8±0.8	400±200	Undetermined
Inflorescences from controlled atmosphere chambers at day 6	2.0±0.06 (ns) ^a^	2100±1200 (ns) ^a^	Undetermined
Open-air inflorescences at day 12	31±19 (*) ^a^	15,000±2800 (****) ^a^	290±110

^a^ One-way ANOVA followed by Tukey’s post hoc test compared to t_0_ results (*p* value < 0.0332 (*), *p* < 0.0001 (****); ns, not significant).

## Data Availability

Data will be made available on request.

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
