# Peer review of "In Pursuit of Optimal Quality: Cultivar-Specific Drying Approaches for Medicinal Cannabis"

_plants, 2024, doi:10.3390/plants13071049_

Round 1

Reviewer 1 Report

Comments and Suggestions for Authors

The paper reports an interesting research and could improve the knowledge about some aspects concerning Cannabis which there are still little investigated in scientific literature. However before publication, it’s necessary give some explanations by the authors and make a revision. 

In the Abbreviations section are reported cannabinoids which are then not taken into account later in the article: CBN, CBNA, CBL, CBLA, CBC; so I recommend deleting them. Also the abbreviations wt% and CFU are missing.

In the Introduction section the authors write at lines 48-49 that dried inflorescence are a primary choice but in recent years this is true only in some Country of the World: I suggest expressly citing the State you are referring to and supporting this statement with a more suitable bibliography.

At Lines 74-76 the authors talk about the classification of Cannabis: it is suggested to express the chemotype as a ratio between the main cannabinoids expressed in terms of neutral cannabinoids or as totals (for example THC TOT/ CBD TOT). At Lines 75-76 it is reported that DW % must be greater than 10 for high variety, but this is not correct because the chemotype is “high” even if the percentage of the main cannabinoid is lower than that value. So 10 must be the value of the THC/CBD ratio and not the percentage of THC in the case of Cannabis high THC.

Since the authors chose to anticipate the Results and Discussion over to Material and Methods section the Tables 1 and 2 are very difficult to understand and follow, so please re-write them indicating the units of measurement of the absolute concentration, the value of the LODs, the number of samples. Moreover, it is essential to anticipate paragraph 3.7 before the table even by inserting it in paragraph 2.1. otherwise it is not possible to understand what is meant by Relative concentration nor are the results obtained clear.

In the Materials and methods section:

in the paragraph 3.1 are reported cannabinoids which are then not taken into account later in the article: so I recommend deleting them. For most of the paper it is not clear the number of samples taken into consideration: only at Line 535 at page 20 the authors are talking about 5 replicates, this data must be anticipated. The authors must explain because they also did not consider several batches of the two chemovars to have more solid data.

In paragraph 3.5 it would be better to include validation data for both the HPLC-PDA and GC/MS methods (even just as tables).

At Line 551 the linearity range seems too wide to me: is this range covered by a unique calibration curve?

Author Response

03.04.24

Comments to the reviewers

We would like to thank the editor and external reviewers for their thoughtful and detailed comments on our paper. We have edited the manuscript to address all of their concerns. We believe that the manuscript is now suitable for publication.

Reviewer 1

  1. In the Abbreviations section are reported cannabinoids which are then not taken into account later in the article: CBN, CBNA, CBL, CBLA, CBC; so I recommend deleting them.

Revised accordingly.

  1. Also the abbreviations wt% and CFU are missing.

We added these abbreviations to the abbreviations section.

3.In the Introduction section the authors write at lines 48-49 that dried inflorescence are a primary choice but in recent years this is true only in some Country of the World: I suggest expressly citing the State you are referring to and supporting this statement with a more suitable bibliography.

We concur with the reviewer's observations and have accordingly revised our statement to read: "Dried inflorescences are among the prevalent medicinal cannabis products accessible to patients (Lines 45-46(."

4.At Lines 74-76 the authors talk about the classification of Cannabis: it is suggested to express the chemotype as a ratio between the main cannabinoids expressed in terms of neutral cannabinoids or as totals (for example THC TOT/ CBD TOT). At Lines 75-76 it is reported that DW % must be greater than 10 for high variety, but this is not correct because the chemotype is “high” even if the percentage of the main cannabinoid is lower than that value. So 10 must be the value of the THC/CBD ratio and not the percentage of THC in the case of Cannabis high THC.

Revised accordingly.

5.Since the authors chose to anticipate the Results and Discussion over to Material and Methods section the Tables 1 and 2 are very difficult to understand and follow, so please re-write them indicating the units of measurement of the absolute concentration, the value of the LODs, the number of samples. Moreover, it is essential to anticipate paragraph 3.7 before the table even by inserting it in paragraph 2.1. otherwise it is not possible to understand what is meant by Relative concentration nor are the results obtained clear.

As the reviewer suggested, we have changed the Tables 1 and 2 to absolute concentrations. We added referral for the relative concentration calculation (page 9, lines 212-213).

6.In the Materials and methods section:in the paragraph 3.1 are reported cannabinoids which are then not taken into account later in the article: so I recommend deleting them. For most of the paper it is not clear the number of samples taken into consideration: only at Line 535 at page 20 the authors are talking about 5 replicates, this data must be anticipated. The authors must explain because they also did not consider several batches of the two chemovars to have more solid data.

As requested, we added the number of samples used for each treatment to the Figures and Tables.

7.In paragraph 3.5 it would be better to include validation data for both the HPLC-PDA and GC/MS methods (even just as tables).

As requested, we added this information to the supplementary section.

8.At Line 551 the linearity range seems too wide to me: is this range covered by a unique calibration curve?

Yes, this range is covered by a unique calibration curve, which is used only for THCA and CBDA (please see Table S1).

On behalf of all co-authors, we thank the reviewers for their comments and valuable suggestions.

Jakob Shimshoni, PhD
Department of Food Science
Institute for Postharvest and Food Sciences
Agricultural Research Organization
Rishon LeZiyyon 
7528809
Israel

Reviewer 2 Report

Comments and Suggestions for Authors

Dear Authors,

The paper I read offered insightful knowledge on the state-of-the-art trend of cultivar-specific drying approaches for medicinal cannabis. This alternative plant gains importance thanks to its numerous beneficial effects on human health.

The proposed manuscript is well written with minor editing of English language required. In Abstract abbrevation to is used without explanation what it represents. Keywords should be written more clearly, namely controlled drying, atmospheric drying. In line 46., family Cannabaceae should be written in italic. Grammarly mistakes should be corrected such as monoterpenes and sesquiterpenes should be written as one word.

Kind regards

Comments on the Quality of English Language

Minor editing of English language required.

Author Response

03.04.24

Comments to the reviewers

We would like to thank the editor and external reviewers for their thoughtful and detailed comments on our paper. We have edited the manuscript to address all of their concerns. We believe that the manuscript is now suitable for publication.

Reviewer 2

1.The proposed manuscript is well written with minor editing of English language required.

A native English speaker went through the manuscript and corrected minor mistakes.

2.In Abstract abbrevation tis used without explanation what it represents.

We have added an explanation as requested.

3.Keywords should be written more clearly, namely controlled drying, atmospheric drying.

Revised as requested.

4.In line 46., family Cannabaceae should be written in italic. Grammarly mistakes should be corrected such as monoterpenes and sesquiterpenes should be written as one word.

We have made the changes throughout the manuscript as requested.

On behalf of all co-authors, we thank the reviewers for their comments and valuable suggestions.

Jakob Shimshoni, PhD
Department of Food Science
Institute for Postharvest and Food Sciences
Agricultural Research Organization
Rishon LeZiyyon 
7528809

Israel

Reviewer 3 Report

Comments and Suggestions for Authors

Comments on the Quality of English Language

The writing English should be improved 

Author Response

Reviewer 3

1.The abstract was too much explicated and you should give a pertinent overview of the work and you should include a paragraph of about 200 words. Please adopt the style of abstract according to the template of plants journal when creating your abstract (the purpose of the study; (2) Methods: briefly describe the main methods or treatments applied; (3) Results: summarize the article’s main findings; (4) Conclusions).

We have revised the abstract accordingly to the journals guidelines, which require a single paragraph abstract without sub-headings.

2.The writing English should be improved by a native English speaker.

A native English speaker went through the manuscript and corrected the spelling mistakes as well as the grammar.

3.The paper is weak discussing, more references need to be added, and try to introduce recent articles. Please eliminate redundancy in the text. For example, the content in lines 28 to 30 duplicates that found in lines 636 to 639. Numerous typos need correction (see attached manuscript). Verify the abbreviation list (e.g., Relative error (RSE) ?). -Keywords: “Cannabis sativa L”. instead of “Cannabis sativa L”. (L not Italic).

Following the recommendation, we have implemented the proposed changes to the manuscript. To this end, we have enriched the discussion section by incorporating a paragraph that compares our findings with those from the existing body of literature on medicinal cannabis drying techniques. This comparison is detailed on page 13 (lines 308-344), page 16 (lines 389-398), and page 19 (lines 479-503), where we compare our controlled atmosphere drying method against several techniques, including microwave drying, hot air drying, infrared drying, and freeze drying, as reported in five different studies published from 2020 to 2023. Furthermore, we have eliminated superfluous text and corrected typographical errors.

  1. If your table spans two pages, it should be labeled as "continued."

We have corrected accordingly.

5.Table 1: Presenting the results of 240 and Gen12 in the same table, would better help to appreciate the behavior of the two chemovars; similarly for Table 2. -Table 4.

All tables now contains the information for both chemovars.

  1. Why you did not determine Mold of the G12 chemovar inflorescences at t0?

We have conducted mold assessments for Gen12, and our findings indicated no mold presence. This detail has now been explicitly stated in the manuscript on page 8, lines 196-198: "Regarding Gen12, the presence of Botrytis cinerea and Alternaria alternata, both at the initial time point (t0) and post-drying, was found to be below the limit of detection."

7.Improve the tables quality.

In response to the feedback, we have enhanced the clarity and conciseness of the tables, making them more straightforward and accessible for interpretation.

  1. The conclusion is too long. Please provide a more concise conclusion.

We have provided a more concise conclusion as requested.

9.Justify the choice of organs analyzed (inflorescences) and the sampling method adopted?

The cannabis inflorescence, which harbors significant concentrations of cannabinoids and terpenes, is the only part of the plant utilized for both medical and recreational purposes. Consequently, it is the focal point of analysis in all pertinent studies within this domain. The sampling method employed in our study aligns with the most widely accepted procedures among researchers and industry practitioners, as evidenced by the literature cited in our manuscript (Upton, R.; ElSohly, M. Cannabis inflorescence: Cannabis spp.; standards of identity, analysis, and quality control; American Herbal Pharmacopoeia, Scotts Valley: 2014; Challa, S.K.R.; Misra, N.; Martynenko, A. Drying of cannabis—State of the practices and future needs. Dry. Technol. 2021, 39, 2055-2064).We have included a paragraph summarizing this points in the methods and materials section as follows: " The cannabis inflorescence, which harbors significant concentrations of cannabinoids and terpenes, is the only part of the plant utilized for both medical and recreational purposes. Consequently, it is the focal point of analysis in all pertinent studies within this domain. The sampling method employed in our study aligns with the most widely accepted procedures among researchers and industry practitioners" (page 20, lines 529-534).

  1. Cannabis is a monoecious species (separate inflorescences): What type of inflorescence (male or female) was analyzed in this study? and was this aspect considered during sampling and analysis?

Commercial medicinal cannabis inflorescences consist exclusively of female flowers, owing to the fact that they contain significantly higher concentrations of cannabinoids compared to their male counterparts. We have included this information in the methods and materials section (page 20, line 522).  

  1. What type of DNA was extracted and sequenced to identify the relative fungal biomass, and why was this choice made???

To estimate the relative fungal biomass, total DNA (encompassing both plant and fungal DNA) was extracted from each specimen. We specifically targeted two predominant pathogens affecting cannabis, Alternaria alternata and Botrytis cinerea, for their quantification as indicators of fungal infection in cannabis, due to their high prevalence in cannabis inflorescence. The calculation of the fungal biomass for each species was performed using species-specific primers targeting the actin housekeeping gene. These values were then normalized to the plant's biomass, utilizing primers specific to the cannabis plant's 18s rRNA gene region. The qPCR technique not only supplements the total CFU counts but also offers enhanced sensitivity for detecting and quantifying the relative presence of specific fungal species. Detailed information regarding this method can be found on pages 21-22, lines 594-629 of the document.

12.Justify the choice of the two chemovars? Why did you select two distinct commercial chemovars? If your aim is to compare the contents of both plants, consider adjusting the title of the article to be more general and encompassing of all the research findings.

Our aim was to investigate if cultivars with distinct cannabinoid and terpene profiles, specifically the high THCA and hybrid cultivars that are widely available in the Israeli market, would respond differently to identical drying conditions. For this purpose, we selected Gen12 (a hybrid cultivar) and chemovar 240 (high in THCA) as representatives of each category. These selections were based on their commercial availability and accessibility to us during the study. Given this context, we now consider that the title accurately represents our research objectives. We added this justification into the methods and materials section (page 20, lines 526-527).

  1. You have worked on the volatile compounds that are major in cannabis, but several studies have shown that this plant is still rich in flavonoids and phenolic acids? In my opinion, you should add a section on flavonoid and phenolic acid contents.

Our analysis did not include flavonoids, which are present in cannabis inflorescences at concentrations lower than cannabinoids and terpenes. Current medical research on cannabis, as well as the medical cannabis industry, predominantly concentrates on cannabinoids and terpenes, attributed to their well-documented and established medicinal properties. Consequently, flavonoids and polyphenols were not considered within the scope of this paper.

  1. Your study explored the impact of varying fast drying conditions on the chemical composition of cannabis inflorescences. Does the sample preservation method, prior to analysis, not influence the quality and yield of metabolites.

In our study, there was no utilization of preservation methods before initiating the drying process or conducting chemical analysis. The drying process commenced immediately on the day of harvest, and chemical analyses were carried out on the same day of harvest and immediately following the conclusion of the drying process. This approach was taken to minimize any potential alterations to the metabolite profile that could arise from preservation techniques, thereby ensuring a direct assessment of the impact of drying conditions on the chemical composition of the cannabis inflorescences.

On behalf of all co-authors, we thank the reviewers for their comments and valuable suggestions.

Jakob Shimshoni, PhD
Department of Food Science
Institute for Postharvest and Food Sciences
Agricultural Research Organization
Rishon LeZiyyon 
7528809

Israel

On behalf of all co-authors, we thank the reviewers for their comments and valuable suggestions.

Jakob Shimshoni, PhD
Department of Food Science
Institute for Postharvest and Food Sciences
Agricultural Research Organization
Rishon LeZiyyon 
7528809

Israel

Round 2

Reviewer 3 Report

Comments and Suggestions for Authors

The manuscript was improved and the authors have responsed to all the comments.